# Multiple episodes of ice loss from the Wilkes Subglacial Basin during the Last Interglacial

Mutsumi Iizuka [1,2,3] ✉, Osamu Seki [1,2] ✉, David J. Wilson [4], Yusuke Suganuma [5,6], Keiji Horikawa [7], Tina van de Flierdt [8], Minoru Ikehara [9], Takuya Itaki [3], Tomohisa Irino [2,10], Masanobu Yamamoto [2,10], Motohiro Hirabayashi [5], Hiroyuki Matsuzaki [11] & Saiko Sugisaki [3]

The Last Interglacial (LIG: 130,000–115,000 years ago) was a period of warmer global mean temperatures and higher and more variable sea levels than the Holocene (11,700–0 years ago). Therefore, a better understanding of Antarctic ice-sheet dynamics during this interval would provide valuable insights for projecting sea-level change in future warming scenarios. Here we present a high-resolution record constraining ice-sheet changes in the Wilkes Subglacial Basin (WSB) of East Antarctica during the LIG, based on analysis of sediment provenance and an ice melt proxy in a marine sediment core retrieved from the Wilkes Land margin. Our sedimentary records, together with existing ice-core records, reveal dynamic fluctuations of the ice sheet in the WSB, with thinning, melting, and potentially retreat leading to ice loss during both early and late stages of the LIG. We suggest that such changes along the East Antarctic Ice Sheet margin may have contributed to fluctuating global sea levels during the LIG.

Global mean sea levels during the Last Interglacial (LIG) are thought to have reached ~6–9 m higher than present, with some records suggesting several peaks within that interval[1–9]. These elevated sea levels are attributed to a combination of ocean thermal expansion, melting of mountain glaciers, and mass loss from the Greenland and Antarctic ice sheets, with at least some contribution from Antarctica probably being required[5,10–12]. Furthermore, in order to explain the multiple sea-level peaks during the LIG, multiple events of ice-sheet mass loss must have occurred. However, detailed records of ice-sheet behaviour during the LIG are lacking, limiting our ability to independently test the proposed global mean sea-level fluctuations

or to resolve their geographic origin. The East Antarctic Ice Sheet stores freshwater with a sea-level equivalent of ~53 m, of which ~19 m resides in marine-based subglacial basins, including the Wilkes, Aurora, and Recovery basins[13]. Because those marine-based sectors are considered to be vulnerable to ocean warming[14], even small oceanographic changes in those regions could have contributed substantially to past fluctuations in global sea level. Whether ice in the marine-based subglacial basins of East Antarctica partially melted during the LIG is critical for understanding future sea-level changes[14], but is not currently resolved by global sea-level records or by local sea-level fingerprints from this interval[5,15]. For this reason, a

[1]Institute of Low Temperature Science, Hokkaido University, Sapporo, Japan. [2]Graduate School of Environmental Science, Hokkaido University, Sapporo, Japan. [3]Geological Survey of Japan, The National Institute of Advanced Industrial Science and Technology (AIST), Tsukuba, Japan. [4]London Geochemistry and Isotope Centre (LOGIC), Institute of Earth and Planetary Sciences, University College London and Birkbeck, University of London, London, UK. [5]National Institute of Polar Research, Tachikawa, Japan. [6]Department of Polar Science, School of Multidisciplinary Sciences, The Graduate University for Advanced Studies (SOKENDAI), Tachikawa, Japan. [7]Faculty of Science, Academic Assembly, University of Toyama, Gofuku, Japan. [8]Department of Earth Science and Engineering, Imperial College London, London, UK. [9]Marine Core Research Institute (MaCRI), Kochi University, Nankoku, Japan. [10]Faculty of Environmental Earth Science, Hokkaido University, Sapporo, Japan. [11]Micro Analysis Laboratory, Tandem accelerator (MALT), The University of Tokyo, Bunkyo, Japan. ✉e-mail: mutsumi.i@pop.lowtem.hokudai.ac.jp; seki@lowtem.hokudai.ac.jp

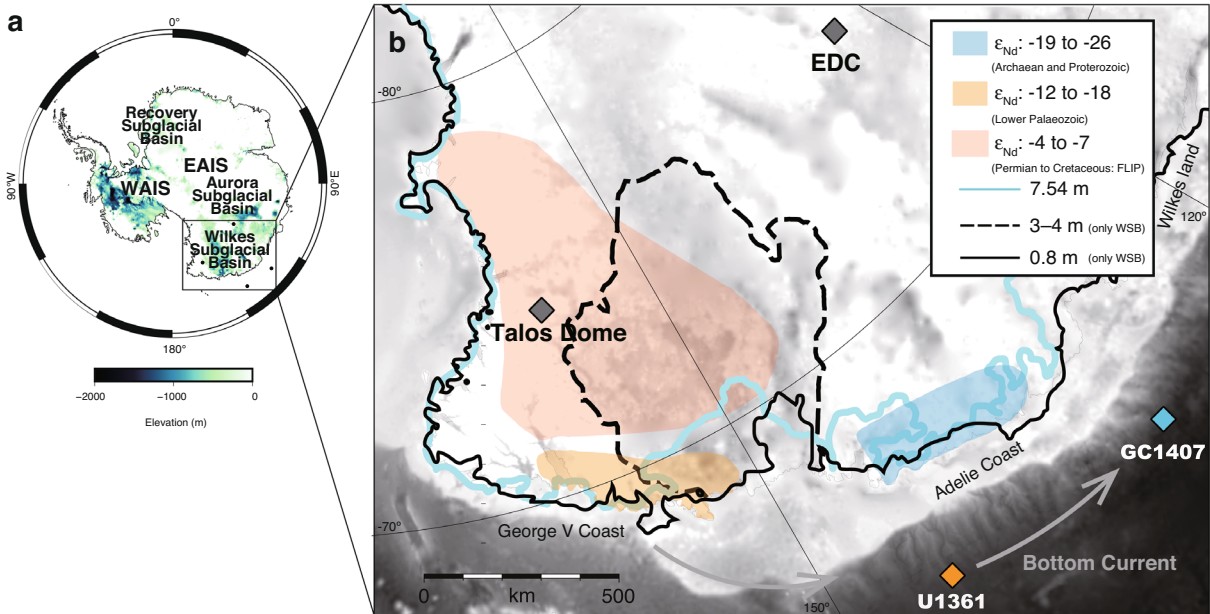

**Fig. 1 | Regional setting of the Wilkes Subglacial Basin. a** Map of Antarctica showing subglacial bedrock elevation below sea level[13] and locations of marine sediment cores and ice cores relevant to this study. Areas in white are above modern sea level and shaded areas are below modern sea level. EAIS, East Antarctic Ice Sheet. WAIS, West Antarctic Ice Sheet. **b** Detailed map of the Wilkes Subglacial Basin (WSB), with grey shading indicating bedrock elevation (as in panel **a**) and coloured shading representing the simplified geographical extent of three geological terranes differentiated according to their neodymium isotope ($\varepsilon_{Nd}$) characteristics (map redrawn from ref. 30). Also shown are the locations of relevant marine sediment cores (GC1407 and U1361A) and ice cores (Talos Dome; EDC,

EPICA Dome C). Lines illustrate positions of the ice-sheet margin in different ice-sheet model simulations and scenarios: black dashed line, fully retreated state of Mengel and Levermann[16]; black solid line, simulated ice margin (8-km resolution and 300-year perturbation) in Sutter et al.[19]; light blue line, maximum simulated Last Interglacial (LIG) retreat in DeConto and Pollard[14]. Sea-level contributions from the retreated states are given in the key. Grey arrow indicates modern flow direction of the bottom currents along the margin. Map created using the Quantarctica GIS package[81] developed by the Norwegian Polar Institute and published under the Creative Commons Attribution 4.0 International License. Map redrawn from ref. 30 with permission.

focus on sedimentary records from the East Antarctic margin is needed.

The ice sheet in the Wilkes Subglacial Basin (WSB) (Fig. 1) has received particular attention because its complete loss would raise global sea levels by ~3–4 m[13,16]. However, when and to what extent the ice-sheet margin in the WSB retreated during the LIG compared to its present position remains a matter of debate. A recent study using sediment provenance tracing in marine sediment core U1361A (64.40°S 143.88°E, 3465 m water depth, Fig. 1b) suggested a significant retreat of the ice-sheet margin in the WSB during several late Pleistocene interglacials, including the LIG[17]. Subsequent studies have argued for a more limited retreat in the WSB during the LIG[18–22], but also suggest a major ice-sheet collapse during Marine Isotope Stage 11 at ~400 thousand years ago (ka)[18]. However, due to limitations in the chronologies and temporal resolution of existing records[17,18], the timing and magnitude of ice-sheet change in the WSB during the LIG remain uncertain.

To better constrain the ice-sheet behaviour in the WSB during the LIG, we generated high-resolution records (300-year resolution) of detrital neodymium (Nd) isotopes and iceberg rafted debris (IBRD), supported by authigenic beryllium (Be) isotopes, in marine sediment core GC1407 (63.75°S, 130.52°E, 3687 m water depth, Fig. 1b). This core was retrieved during a survey of the Antarctic Geological and Geophysical Research Project by the Japan National Oil Corporation in 1993. Neodymium isotope ratios are expressed as $\varepsilon_{Nd}$ values, the deviation of $^{143}Nd/^{144}Nd$ ratios from the Chondritic Uniform Reservoir value in parts per 10,000, and provide evidence on the locus of subglacial erosion[17] (see Methods). Authigenic Be isotope ratios are reported as $^{10}Be/^{9}Be$ ratios and are considered to be an indicator of meltwater inputs[23–25] (see Methods). In addition, to evaluate the extent of ice-sheet margin retreat in the WSB, we estimated changes in

elevation at Talos Dome during the LIG based on the reported oxygen isotope ($\delta^{18}O$) records in Antarctic ice cores[22]. The chronologies for marine sediment cores GC1407 and U1361A are tuned to the Antarctic Ice Core Chronology 2012 (AICC2012)[26,27], based on the onset and end of the LIG (see Methods).

## Results and discussion
### Two episodes of provenance change during the LIG
Detrital sediment Nd isotopes have been widely applied as an indicator of sediment provenance around Antarctica[17,28–30]. The detrital record of fine-grained sediment (<63 μm) in core GC1407 shows relatively low $\varepsilon_{Nd}$ values (around −16 to −15) during the Holocene and relatively higher values with more variability (−17 to −13) during the LIG (Fig. 2d, i). Two episodic increases in $\varepsilon_{Nd}$ values (reaching −14 to −13) are recognised within the LIG (129–126 ka and 122–118 ka) (Fig. 2i), suggesting dynamic changes in sediment provenance. Interestingly, the updated age model for core U1361A also reveals some evidence for two Nd isotope peaks during the LIG in core U1361A, although they are defined by only a small number of data points (Fig. 2i). Given the good agreement in the timing of the two peaks between the two cores, our high-resolution record from core GC1407 appears to reinforce the presence of two peaks in core U1361A. Although there is a clear offset of ~4 $\varepsilon_{Nd}$ units between the two records (Fig. 2d, i), the similarity in terms of the magnitude of changes and temporal patterns suggests that the two events of increasing $\varepsilon_{Nd}$ values during the LIG likely reflect the same phenomenon of provenance change.

Sediment provenance variations indicated by the Nd isotope record in core U1361A are thought to reflect the waxing and waning of the ice-sheet margin in the WSB[17]. This interpretation is based upon the ultimate source of detrital sediment input to marine sequences on the

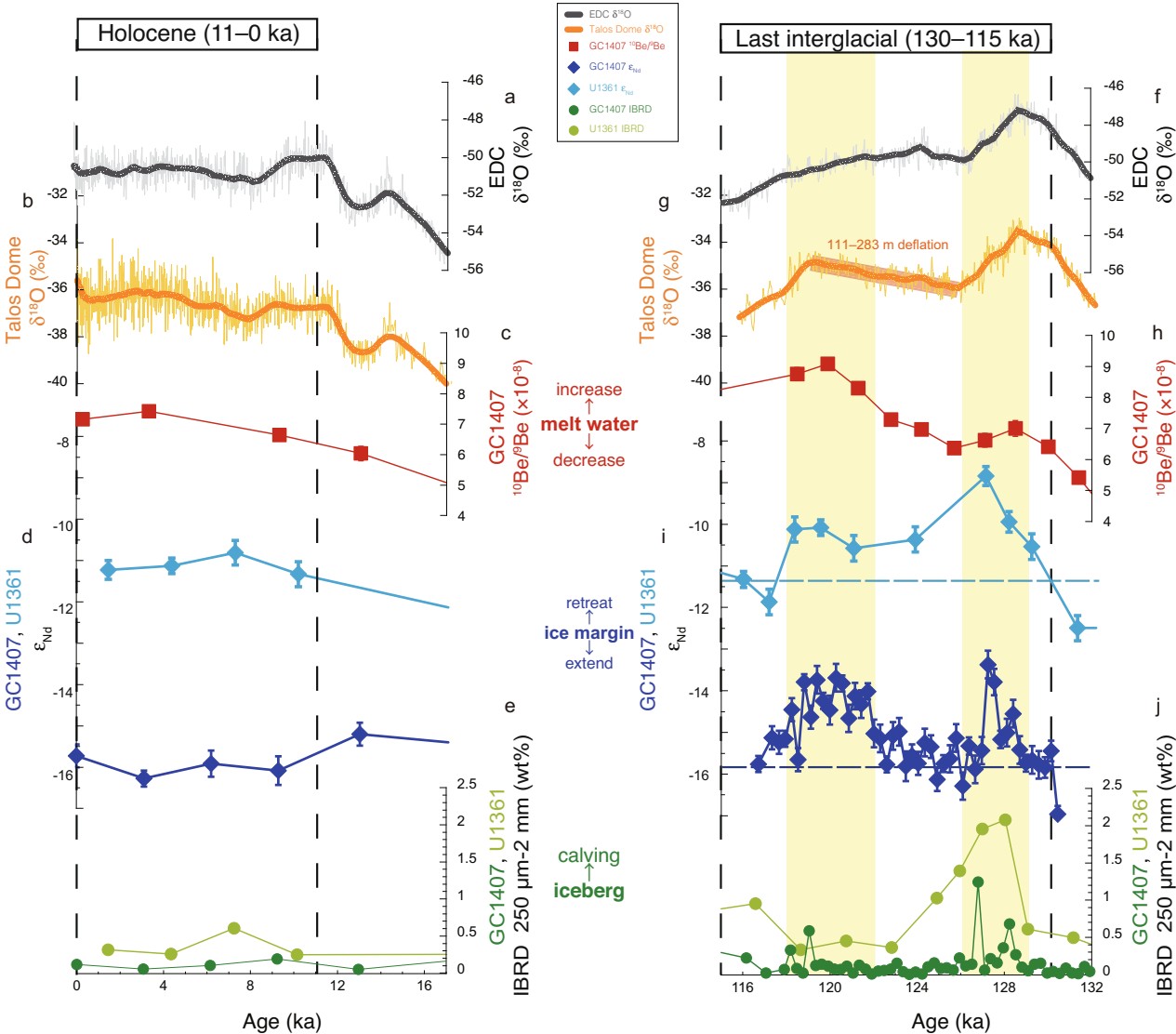

**Fig. 2 | Multi-proxy evidence for variability in the Wilkes Subglacial Basin during the Holocene and Last Interglacial. a–e** and **f–j** Multi-proxy data for the Holocene and Last Interglacial (LIG), respectively. **a, f** δ18O records from EPICA Dome C (EDC) ice core (thin and bold black lines are the raw signals and weighted smoothing, respectively)[39]. **b, g** δ18O records from Talos Dome ice core (thin and bold orange lines are the raw signals and weighted smoothing, respectively)[39]. **c, h** Authigenic 10Be/9Be ratios from core GC1407 (red squares, this study) (error

bars are 1 s.d.). **d, i** Detrital sediment Nd isotopes ($\varepsilon_{Nd}$) from core GC1407 (dark blue diamonds, this study) and core U1361A (light blue diamonds)[17] (error bars are 2 s.d.). **e, j** Iceberg rafted debris (IBRD, percent 250 μm–2 mm) from core GC1407 (dark green circles, this study) and core U1361A (light green circles)[17]. Light blue and dark blue dashed lines in i indicate the core top $\varepsilon_{Nd}$ value of U1361A and GC1407, respectively. Yellow bars indicate two episodes of inferred Wilkes Sub-glacial Basin ice mass loss during the early and late LIG.

Antarctic margin being from the erosion of proximal and/or upstream coastal bedrock[31–36], which can be expected to change with advance or retreat of the ice-sheet margin. Furthermore, the $\varepsilon_{Nd}$ values of the bedrock vary by region[17,28–30] (Fig. 1b). The coastal bedrock in the vicinity of core GC1407 (Adelie Coast; around 145–135°E) has $\varepsilon_{Nd}$ values of −19 to −26 (Archaean and Proterozoic age), while coastal bedrock proximal to core U1361A (George V Coast; around 150–160°E) has $\varepsilon_{Nd}$ values of −12 to −18 (Lower Palaeozoic age)[30]. In contrast, $\varepsilon_{Nd}$ values in the WSB, inland of George V Coast, are inferred to be more radiogenic (−4 to −7), reflecting the Permian-Cretaceous rocks of the Ferrar Large Igneous Province (FLIP) and associated Beacon Supergroup[30,37]. At present, detrital sediments are dominantly trans-ported from the coastal margin of the WSB to the location of core U1361A[17]. Since the LIG was also an interglacial climate state, it can be assumed that the sediment transport regime during the LIG was broadly similar to the modern day. Therefore, the high $\varepsilon_{Nd}$ values in core U1361A during the LIG (Fig. 2i) are interpreted to reflect the

supply of detritus from inland bedrock to the continental rise due to WSB ice-sheet retreat[17].

Since sediments are transported westwards by bottom currents (Antarctic Bottom Water)[35,36] near Wilkes Land, the detrital Nd isotope record in core GC1407 likely reflects the erosion of coastal bedrock upstream of the site (i.e. Adelie Coast and George V Coast) (Fig. 1b), especially during interglacials when the bottom current intensified[35,36] (see Methods). If retreat of the ice-sheet margin in the WSB enhanced the supply of sediments from inland FLIP sources, relatively high detrital $\varepsilon_{Nd}$ values would also be expected in core GC1407, whereas the supply of detritus from the Adelie Coast would result in a lowering of $\varepsilon_{Nd}$ values. As such, we suggest that the relatively low $\varepsilon_{Nd}$ values in core GC1407 compared to core U1361A (-4 $\varepsilon_{Nd}$ units offset) probably indi-cate continuous detrital input from the Adelie Coast through the Holocene and LIG (Fig. 2d, i). On the other hand, we suggest that the two Nd isotope peaks observed in core GC1407 during the LIG, coin-cident with peaks in core U1361A, reflect ice-sheet margin retreat

events in the WSB. Overall, our new high-resolution record, together with improved constraints on the chronology of core U1361A, reveals that ice-sheet margin retreat in the WSB occurred during both the early and late LIG. Although our record does not rule out the possibility of ice-sheet margin retreat in the Adelie Coast region, several independent ice-sheet modelling studies suggest that ice-sheet margin retreat in this region would have been limited during the LIG[14,19] (Fig. 1b), which is also consistent with observations over recent decades from the Adelie Coast[38].

## Extent of ice-sheet melting in the WSB

To provide further evidence on the nature and magnitude of ice-sheet margin changes in the WSB that are implied by the two peaks in the detrital Nd isotope record, we measured authigenic Be isotopes in core GC1407. Changes in the $^{10}Be/^9Be$ ratio in marine sediments proximal to ice-sheet margins can be used to infer regional inputs from ice melt[23–25]. The production of $^{10}Be$ in the atmosphere is followed by its deposition onto the ocean and ice sheets, such that the melting of ice sheets and icebergs releases the accumulated $^{10}Be$ into the ocean, where it is scavenged by particles and accumulates in the authigenic fraction of marine sediments[23]. In contrast, $^9Be$ is the stable naturally-occurring isotope found in bedrock and supplied via erosion and weathering. An increase in the $^{10}Be/^9Be$ ratio in marine sediments can therefore be interpreted as indicating a significant ice melting event[23] (see Methods).

The $^{10}Be/^9Be$ record in core GC1407 provides strong support for ice mass loss from the WSB during the LIG. The $^{10}Be/^9Be$ ratios vary from $6.0 \times 10^{-9}$ to $9.1 \times 10^{-9}$ during the LIG, with a two-stepped increase leading to maximum values at ~120 ka (Fig. 2h). This finding points to an increase in ice melting during the late LIG, and suggests that melting was more pronounced from 122–118 ka than from 129–126 ka. The two-stepped increase in $^{10}Be/^9Be$ ratios also coincides with the Nd isotope peaks (Fig. 2h, i), supporting the interpretation of those provenance shifts as reflecting ice-sheet retreat rather than reflecting changes in other processes such as deep-current sediment transport.

The content of iceberg rafted debris (IBRD; weight percent of grains of 250 μm–2 mm diameter), which provides a qualitative indicator of iceberg discharge[17], was also measured in core GC1407 to infer the causes of ice mass loss from the WSB. Peaks in IBRD in cores GC1407 and U1361 are observed during the early LIG and coincide in timing (Fig. 2j), which suggests that iceberg calving was an important process for ice mass loss during the early LIG. On the other hand, the $^{10}Be/^9Be$ record reaches its maximum during the late LIG, indicating that ice-shelf basal melting[23] may have played a relatively more important role than calving in ice mass loss from the WSB during the late LIG.

## Assessment of ice mass loss in the WSB

To further evaluate the ice-sheet retreat in the WSB during the late LIG as indicated by our sedimentary records, we made a comparison to the δ18O records from the Talos Dome and EPICA Dome C (EDC) ice cores[39]. Differences in the δ18O records between the Talos Dome and EDC ice cores have been proposed to provide an indicator for the advance and retreat of the ice-sheet margin in the WSB[19,22]. Because the Talos Dome site is located in the headlands of the WSB outflow region, it is sensitive to changes of the ice-sheet margin in the WSB[19,22], unlike inland plateau sites such as EDC (Fig. 1b). Therefore, the δ18O record from the Talos Dome ice core has been used to estimate temperature increases that were associated with ice-sheet elevation drops, which can be linked to regional ice-sheet mass loss[19,21,22,40–42].

During the early LIG (130–126 ka), the δ18O records at the Talos Dome and EDC sites vary in parallel (Fig. 2f, g). In contrast, the δ18O record at the Talos Dome increases by 1.5‰ from 126 ka to 118 ka, whereas the EDC record shows the opposite trend with a decrease during this period (Fig. 2f, g)[39]. The increase in the δ18O record at Talos

Dome compared to that of EDC is interpreted as a site temperature increase caused by a decrease in site elevation[22]. Notably, the 1.5‰ increase in δ18O values at Talos Dome during the late LIG coincides with the increases in $\varepsilon_{Nd}$ values and $^{10}Be/^9Be$ ratios in cores GC1407 and U1361 (Fig. 2). This temporal agreement allows us to link ice thinning at the Talos Dome site to ice-sheet margin retreat and melting in the WSB, as inferred from the offshore records. The timing of these changes in the ice core record further supports the hypothesis that ice-sheet retreat and melting may have been more pronounced during the late LIG (122–118 ka) than during the early LIG (129–126 ka).

Following Crotti et al. (2022)[22], we estimate the change in elevation at Talos Dome during the LIG by applying three different δ18O-elevation relationships derived from climate simulations and observations (−0.53‰/100 m, −0.93‰/100 m, and −1.35‰/100 m)[21,22,43,44]. Based on those three different relationships, the 1.5‰ increase in δ18O values at Talos Dome is equivalent to a drop in local ice-sheet surface elevation of 111–283 m. Although their study focused on the early LIG, Sutter et al. (2020)[19] presented a model in which the elevation drop of 100–200 m at Talos Dome could be related to an inland retreat of the ice margin in the WSB by several hundred km (their experiment with 8-km resolution, 300-year perturbation)[19] (black line in Fig. 1b). Another recent model presented by Crotti et al. (2022)[22] also indicates ice margin retreat in the WSB of ~100 km during the 133–115 ka interval, and notably the greatest retreat occurred towards the end of the LIG, which is consistent with our interpretations. In summary, the above findings provide multiple lines of evidence from which we infer ice mass loss from the WSB during the late LIG from 122–118 ka.

## Drivers of ice mass loss in the WSB

Previous studies of Antarctic ice-sheet variations during the LIG have focused on the early LIG (~128 ka) when air temperature rise in Antarctica reached its peak[19,21]. However, our records reveal evidence for extensive ice mass loss from the WSB during the late LIG (~120 ka), when air temperatures were comparable to the present (Fig. 3a). Hence, atmospheric forcing was unlikely to have been a trigger for ice mass loss during the late LIG. Instead, since the ice sheet in the WSB is marine-based[13], ocean forcing from changes in ocean circulation and/or ocean warming may have played critical roles in the ice-sheet mass loss during the late LIG.

In general, the LIG was characterised by sea surface temperatures (SST) in the Southern Ocean that were approximately 2 °C warmer than the pre-industrial Holocene[45] and by a southward-shifted Polar Front in the Indian Ocean[46]. Both these conditions could have contributed to enhanced ocean forcing of ice-sheet mass loss, with the frontal shifts potentially helping to drive incursions of relatively warm Circumpolar Deep Water onto the continental shelves, which could lead to inland migration of grounding lines[47,48]. Notably, deep-ocean temperatures in the Southern Ocean also remained at elevated values from the early to the late LIG[49] (Fig. 3b), pointing to warmer than present conditions in the sub-surface Southern Ocean even during the late LIG. Furthermore, while the spatial coverage is sparse, two marked warming peaks can be resolved in some SST reconstructions from the Southern Ocean, with one during the early LIG and another during the late LIG[50,51] (Fig. 3c). Although the above records are from lower latitudes of the Southern Ocean than the Antarctic margin, recent diatom-based SST reconstructions from close to the margin also indicate warmer conditions throughout the LIG than today[46]. Overall, these records are consistent with ocean forcing as the primary driver of ice loss from the ice sheet in the WSB during the LIG, although the exact mechanisms responsible for the ocean warming remain to be fully explored.

## Contribution of ice loss from the WSB to global sea-level rise

Our data provide a unique opportunity to compare a well-resolved ice-proximal record of changes in the East Antarctic Ice Sheet to global

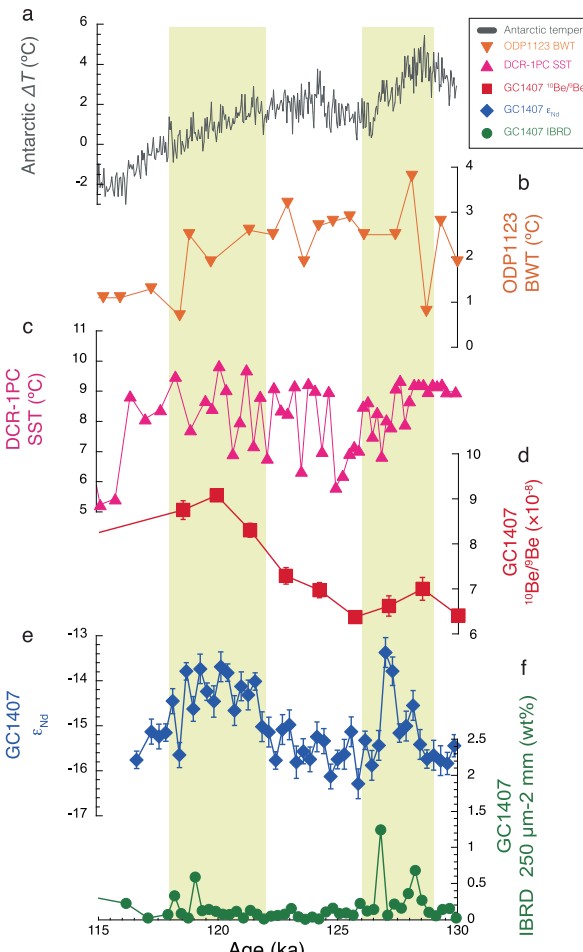

**Fig. 3 | Comparison of core GC1407 records to regional palaeoclimate records.**
**a** Antarctic ice-core temperature difference ($\Delta T$, difference from mean values of
the last millennium) derived from deuterium isotopes at EPICA Dome C (EDC)[39]
plotted on the AICC2012 age scale (grey line). **b** Southern Ocean bottom water
temperature (BWT) from benthic foraminiferal Mg/Ca ratios at Ocean Drilling
Program (ODP) Site 1123 (orange triangles)[49]. **c** Southern Ocean sea surface
temperature (SST) from diatom assemblages in core DCR-1PC (pink triangles)[50].
**d** Authigenic $^{10}Be/^{9}Be$ ratios from core GC1407 (red squares, this study) (error
bars are 1 s.d.). **e** Detrital sediment Nd isotopes ($\varepsilon_{Nd}$) from core GC1407 (dark
blue diamonds, this study) (error bars are 2 s.d.). **f** Iceberg rafted debris (IBRD,
percent 250 μm–2 mm) from core GC1407 (green circles, this study). Yellow bars
indicate two episodes of inferred Wilkes Subglacial Basin ice mass loss during
the early and late Last Interglacial.

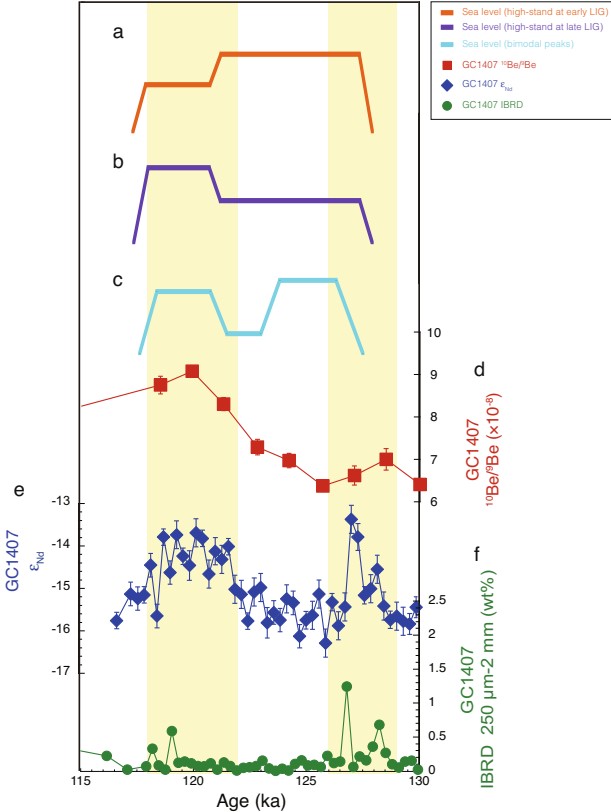

**Fig. 4 | Comparison of core GC1407 records to sea-level scenarios during the
Last Interglacial. a–c** Simplified eustatic sea-level records from a range of global
sites[1–4, 6–9]. **d** Authigenic $^{10}Be/^{9}Be$ ratios from core GC1407 (red squares, this study)
(error bars are 1 s.d.). **e** Detrital sediment Nd isotopes ($\varepsilon_{Nd}$) from core GC1407 (dark
blue diamonds, this study) (error bars are 2 s.d.). **f** Iceberg rafted debris (IBRD,
percent 250 μm–2 mm) from core GC1407 (green circles, this study). Yellow bars
indicate two episodes of inferred Wilkes Subglacial Basin ice mass loss during
the early and late Last Interglacial.

sea-level records for this interglacial period (Fig. 4)[1–9]. Sea-level varia-
bility during the LIG remains uncertain, but three broad scenarios have
been proposed: (1) a high-stand during the early LIG[8,9]; (2) a high-stand
for the initial several thousand years followed by a secondary rise
around 120 ka[1,6]; and (3) bimodal sea-level peaks during the early and
late LIG[2–4] (Fig. 4a–c). Recent studies have suggested that parts of the
West Antarctic Ice Sheet were lost during the early LIG and could have
made the main contribution to global sea-level rise at that time[20,52].
However, the causes of a possible sea-level rise during the late LIG for
scenarios 2 and 3 remain unknown. Our study provides evidence for
ice mass loss from the WSB during the late LIG, thereby supporting an
Antarctic origin for global sea-level rise at this time, which would be
consistent with sea-level rise scenarios 2 and 3. Taken together with
previous studies[11,52], we suggest that ice sheets in different regions of
Antarctica lost ice mass in a step-wise pattern during the LIG, which
may have contributed to dynamic fluctuations of global sea level in any
of the scenarios 1–3 during the LIG.

Ice-sheet model experiments[19,22] indicate that a 100–200 m drop
in elevation at Talos Dome due to WSB ice-sheet margin retreat would
translate to a sea-level rise of ~0.4–0.8 m. Combining those model
results with our study, we suggest that ice-sheet mass loss in the WSB
probably contributed to a similar magnitude of sea-level rise during
the late LIG. Hence, such a change could only explain a small portion of
the global sea-level rise of ~6–9 m estimated for the LIG[5]. If the global
sea-level rise was closer to the maximum estimates (~9 m)[5], then sig-
nificant ice mass loss from other sectors of East Antarctica would still
be required, after accounting for contributions from mass loss of the
marine-based West Antarctic Ice Sheet (~3.3 m) and a portion of the
Greenland Ice Sheet (likely up to ~2 m)[11,53–55]. Within East Antarctica, the
ice sheets in the Aurora and Recovery Subglacial Basins (Fig. 1a)
are also thought to be vulnerable to ocean warming[56,57], and represent
likely candidates for additional global sea-level rise during the LIG.
Comparable high-resolution records from the different sectors of the
East Antarctic Ice Sheet will therefore be required to better constrain
the ice-sheet dynamics and the origin of sea-level variability during
the LIG.

## Methods
### Material
Gravity core GC1407 was collected from the continental rise offshore
of Wilkes Land (63.75°S, 130.52°E, 3687 m water depth) (Fig. 1) during a
survey of the Antarctic Geological and Geophysical Research Project
(FY1993) by the Japan National Oil Corporation. The total length of

core GC1407 is 5.35 m. Glacial-interglacial cycles are well resolved in the core, with glacial intervals indicated by silty-clay with occasional laminations, and interglacial intervals marked by siliceous silty-clay with high $SiO_2$ content.

## Detrital neodymium isotope measurements

To measure detrital Nd isotopes in core GC1407, ~200 mg of the bulk fine-grained (<63 μm) sediment fraction was first leached by sequential leaching methods[58]. The Fe-Mn oxide fraction was leached using 0.05 M hydroxylamine hydrochloride (HH) in 25% acetic acid (Ultra-pure from Kanto Chemical Co., INC., Tokyo, Japan) in a water bath at 85 °C for four hours. Opal was then leached using 1 M NaOH in a water bath at 80 °C for 30 min. After those procedures, the leached samples were rinsed in ultrapure water three times and dried in an oven (50 °C) for ~24 h. Then, ~60 mg samples were completely digested with $HNO_3$-$HF$-$HClO_4$ at 130 °C for at least one day.

Chemical separation of Nd was carried out as follows: (i) major elements were removed from the sample solution using a cation exchange resin column (MCI GEL CK08P, Mitsubishi Chemical Corporation) with 1.8 M HCl; (ii) the REE fraction was extracted using 4.5 M HCl eluent; and (iii) the Nd fraction was isolated using a Ln-spec resin column (50–100 μm, Eichrom Technologies) with 0.25 M HCl eluent. All reagents were trace metal grade reagents (TAMAPURE-AA-100 from Tama Chemicals, Ltd., Tokyo, Japan) and all procedures were conducted in a clean laboratory at the University of Toyama.

Samples were then dissolved in 0.16 M or 0.1 M $HNO_3$ and ultra-sonicated for 1 h prior to analysis using either a first generation (Nu Plasma) or second generation (Nu Plasma II) Nu Instruments HR MC-ICP-MS in the MAGIC laboratories at Imperial College London. The sample introduction system consisted of a Glass Expansion MicroMist nebuliser (typical uptake rate ~100 μL/min) connected to a DSN-100 desolvation nebuliser system. Measurements consisted of 50 cycles. All Nd isotope ratios were corrected for instrumental mass bias using the exponential law and a $^{146}Nd/^{144}Nd$ ratio of 0.7219. The reported $^{143}Nd/^{144}Nd$ ratios were corrected to a JNdi-1 $^{143}Nd/^{144}Nd$ ratio of 0.512115 (ref. 59) using bracketing standards. A correction for $^{144}Sm$ interference was applied, but all samples fell significantly below the threshold for accurate correction of $^{144}Sm$ on the $^{144}Nd$ signal (i.e. <1000 ppm). The external reproducibility of sample data was estimated from the within-session standard deviation (2 s.d.) of the JNdi-1 standards, while its accuracy is demonstrated by repeat analyses of rock standard BCR-2, which yielded $^{143}Nd/^{144}Nd = 0.512641 \pm 0.000015$ ($n = 16$), in excellent agreement with the literature value[60].

## Sortable silt measurements

In preparation for grain-size analysis[61], carbonate and biogenic silica were removed from sediment samples by reacting with 1 M acetic acid for 24 h, followed by heating to 85 °C in 2 M sodium carbonate for 5 h, with samples being agitated several times during each step. Samples were then suspended in 0.2% sodium diphosphate decahydrate solution and placed on a rotating wheel before analysis using a laser diffraction-scattering particle size analyser (Horiba LA-920). Repeated analysis was performed on a subset of samples in an arbitrary order over several days and the mean standard deviation of replicate analyses was ±0.1 μm.

## Neodymium isotope records in core GC1407 as a provenance tracer

The detrital Nd isotope record in core GC1407 (<63 μm fraction) shows large variations in $\varepsilon_{Nd}$ values from −17 to −13 during the last glacial-interglacial cycle (Supplementary Fig. 1b). In general, relatively low values were recorded during interglacial periods, with higher values during glacial periods. Fine-grained sediment (<63 μm) at Antarctic continental rise sites such as GC1407 is predominantly composed of contourite and turbidite deposits (e.g., ref. 32). Therefore, the

long-term glacial-interglacial trends in Nd isotopes probably reflect changes in the relative contribution of detrital sources among contourites and turbidites.

The contourites are supplied from the upstream region by the bottom water current. In core GC1407, the mean grain size of the sortable silt fraction (10−63 μm) ($\overline{SS}$), which is indicative of bottom water current intensity[61], shows large glacial-interglacial variations, with finer grain size during glacials and coarser grain size during interglacials (Supplementary Fig. 1). This observation suggests that bottom current flow was intensified during the interglacial periods, which would have led to enhanced contourite supply from the upstream region. Furthermore, the absence of turbidite layers in core GC1407 during the LIG suggests that contourites were the predominant source throughout the LIG. Therefore, the fluctuations of detrital Nd isotope composition during the LIG should reflect changes in the sediment provenance upstream of the site.

## Iceberg rafted debris (IBRD) measurements

Samples were processed for grain size analysis at Hokkaido University to provide evidence on the IBRD content. Dried samples were weighed and then wet-sieved to recover the coarse sand fraction (percent 250 μm−2 mm), which has previously been used to indicate IBRD in Antarctic studies[17]. The 250 μm−2 mm fraction was weighed to obtain a weight percent of IBRD (percent 250 μm−2 mm) relative to the bulk sediment. For this study, we did not calculate IBRD mass accumulation rates, but they would scale almost directly with the weight percent IBRD (since our age model assumes constant sedimentation rates in the LIG and there are no large shifts or trends in dry bulk density in this interval).

## *Cycladophora davisiana* relative abundance counts

Freeze-dried subsamples were weighed (approximately 1−2 g) and then treated with 15% $H_2O_2$ to remove the organic matter, followed by treatment with HCl solution to remove the calcium carbonate. The samples were then wet sieved (45 μm mesh size), after which two types of permanent slides were made from the residue to quantify the abundance (Q-slide) and for faunal analysis (F-slide)[62]. Briefly, to prepare the Q-slides, all residues were transferred to a 200 mL beaker containing 100 mL of distilled water. The solution was then well mixed, after which a 500 μL sample was taken from the suspension using a micropipette and dropped onto a glass slide. The sample was then dried and mounted with Norland optical adhesive. The F-slides were made from the remaining residues in the beaker and then mounted with Norland. On average, 100 individuals were counted on each slide, and the abundance of *C. davisiana* is reported in percent as the number per 100 individuals.

## XRF core-scanning measurements

The $SiO_2$ content was analysed by X-ray fluorescence (XRF) analysis of slab samples conducted at 1-cm intervals using the TATSCAN-F2[63] at Kochi University. Samples were wrapped in thin prolene film and scanned with a scanning diameter of 1 cm.

## Authigenic beryllium isotope measurements

Samples for Be isotope analysis were prepared at the National Institute of Polar Research according to the chemical procedure established by ref. 64 and revised by ref. 65. Authigenic Be was extracted from 2 g dry samples by soaking them in 20 ml of leaching solution (0.04 M hydroxylamine hydrochloride and 25% acetic acid) at 95 ± 5 °C for 6 h. A 2 ml aliquot of the resulting leachate was sampled for measurement of the natural $^9Be$ concentration using inductively coupled plasma mass spectrometry (ICP-MS). The remaining solution was spiked with 300 μl of $^9Be$ carrier ($9.833 \times 10^{-4}$ g/g $^9Be$) before Be purification by chromatography in order to determine accurate $^{10}Be$ concentrations using accelerator mass spectrometer (AMS) measurements of $^{10}Be/^9Be$

ratios at Micro Analysis Laboratory, Tandem accelerator, University of Tokyo (MALT)[66,67]. Sample [10]Be concentrations were calculated from the [10]Be/[9]Be ratios normalized to the KN standard with a nominal [10]Be/[9]Be ratio of $8.56 \times 10^{-12}$ (ref. 68), and the reported authigenic [10]Be concentrations were decay-corrected using the [10]Be half-life of $1.387 \pm 0.012$ Ma[69,70]. Uncertainties on the authigenic [10]Be/[9]Be ratios were calculated by propagating the uncertainties on the ICP-MS and AMS measurements.

## Use of beryllium isotopes in core GC1407 as a meltwater proxy

Beryllium isotopes have been used as tracers in various fields of earth sciences, providing proxies for geomagnetic field strength[65], sea-ice extent, ice-shelf coverage[25,71], and freshwater discharge from melting ice[23,65]. Cosmogenic [10]Be is produced in the upper atmosphere by the interaction of cosmic rays and is globally homogenised, with an atmospheric residence time of 1–2 years, before being deposited as meteoric [10]Be on the Earth surface[72]. Although cosmic rays also interact with surface rocks and produce in-situ [10]Be, the production rates for this process are orders of magnitude lower than for meteoric [10]Be. In contrast, stable [9]Be occurs naturally as a trace component of crustal rocks and is transported by glacial erosion and weathering to the Antarctic coastal margin[25]. Therefore, [10]Be/[9]Be ratios can be applied in Antarctic margin sediments to constrain sedimentary processes and paleoenvironments, including freshwater supply during events of large-scale ice-sheet retreat. Since using the [10]Be/[9]Be ratio corrects for variations related to particle size, retention behaviour, and dilution, this ratio has been proposed as a more effective meltwater proxy than using [10]Be concentrations alone[23–25].

Authigenic [10]Be and [9]Be concentrations in core GC1407 show similar long-term changes, with higher values during interglacial periods (Holocene and LIG) and lower values during glacial periods (Marine Isotope Stages 2 and 6) (Supplementary Fig. 2). Most notably, authigenic [10]Be concentrations during glacial periods decrease to ~10% of their value during interglacial periods, indicating a glacial-interglacial climate modulation of the records rather than a geomagnetic control[23,65]. Because the area of sea-ice cover changed significantly through glacial-interglacial cycles, the major decrease in [10]Be inputs during glacial periods likely reflects expanded and/or more permanent sea-ice coverage or the presence of ice shelves over the site. In contrast, variations in authigenic [10]Be inputs within interglacial periods cannot be attributed to sea-ice changes because the site of core GC1407 was not covered by sea ice (or ice shelves) during the LIG[73]. Hence, the [10]Be and [10]Be/[9]Be variations in core GC1407 during the LIG likely reflect variations in the excess supply of [10]Be to this region via freshwater input from melting ice. Rapid scavenging by particles in the upper water column would transport the [10]Be to depth, enabling it to accumulate in the authigenic fraction of the marine sediments. Although other factors such as the scavenging of [10]Be by biogenic opal could potentially influence the deep-ocean sedimentary record of [10]Be, the $SiO_2$ content in core GC1407 does not show significant fluctuations during the LIG (Supplementary Fig. 3d), which suggests that any such effect is probably limited in core GC1407. Hence, similar to previous studies (e.g., refs. 23,65), we use the authigenic [10]Be/[9]Be record in core GC1407 as an indicator of regional freshwater inputs from melting ice.

## Correlation between marine sediment cores

Benthic foraminiferal $\delta^{18}O$ records have been extensively applied to determine chronologies for marine sediment sequences, but benthic foraminifera are not continuously preserved in cores GC1407 or U1361A. Therefore, we have identified the interval of the LIG in cores GC1407 and U1361A based on records of *C. davisiana* and productivity that have previously been used to determine chronologies in the Southern Ocean (e.g., refs. 74,75).

First, we identified the interval of the LIG in core GC1407 based on the correlation of *C. davisiana* stratigraphy between cores GC1407 and

RC11-120 (43.52°S, 79.87°E; 3135 m), since the chronology of RC11-120 has been well constrained using its planktonic foraminiferal (*Globigerina bulloides*) $\delta^{18}O$ record[76,77] (Supplementary Figs. 3 and 4). The relative abundance of *C. davisiana* fluctuates through glacial-interglacial cycles, with high contents during interglacials and low contents during glacials in most of the Southern Ocean[74,78]. As shown in Supplementary Fig. 3, temporal variations of *C. davisiana* in cores GC1407 and RC11-120 are similar over the last 140 kyr, which provides two tie points for core GC1407 near the start and end of the LIG (Supplementary Data).

Second, the LIG interval in core U1361A was constrained based on correlation of the productivity records from cores U1361A[17] and GC1407. In the Southern Ocean, the glacial-interglacial productivity pattern depends on latitude, with low and high productivity during glacial and interglacial periods, respectively, in the high latitudes of the Indian sector of the Southern Ocean[75]. Given the proximity of cores GC1407 and U1361A, their productivity patterns are expected to be similar through time. Indeed, the $SiO_2$ content in core GC1407 and Ba/Al ratios in core U1361A (proposed as a productivity proxy in the study area[17]) show similar patterns on orbital to millennial timescales, including during the LIG interval (Supplementary Figs. 3 and 4). The chronology of core U1361A was therefore aligned with core GC1407 at two tie points based on their productivity records (Supplementary Figs. 3 and 4 and Supplementary Data).

## Age control for marine sediment cores and ice cores

To compare the marine sediment core and ice core data, we transferred the Antarctic Ice Core Chronology 2012 (AICC2012)[26,27] to the marine sediment core records. Recent studies have proposed that the correlation between sea surface temperature (SST) in the Southern Ocean and air temperature proxies ($\delta D$ or $\delta^{18}O$) in the Antarctic ice cores can be used to tune the age scales of marine sediment cores, including during the LIG when there is a strong similarity between the marine and ice core records[50,79]. However, SST records are not available for cores GC1407 or U1361A. Therefore, the planktonic foraminiferal $\delta^{18}O$ record in core RC11-120, which is taken as an indicator of SST, and the $\delta D$ record from the Dome Fuji ice core[80] were compared in order to transfer the age of core RC11-120 onto the AICC2012 age scale, using three tie points (Supplementary Fig. 4, Supplementary Data). Then, this chronology was transferred to core GC1407 based on the correlation of the *C. davisiana* records between cores RC11-120 and GC1407, as described above. Finally, this chronology was transferred to core U1361A based on the correlation between the productivity records from cores GC1407 and U1361A, as also described above. With these age models, the 2.8–3.8 m depth interval in core GC1407 and the 1.4–2.1 m depth interval in core U1361A correspond approximately to the LIG (Supplementary Fig. 3). The robustness of the chronology for core GC1407 is also supported by the last appearance of the diatom *Hemidiscus karstenii*, which is identified at 4 m depth in this core, and is known to occur within late Marine Isotope Stage 6[50]. Note that these age models imply significantly higher sedimentation rates during the LIG than during the glacial intervals (around 3.5 to 5 times higher) (Supplementary Fig. 5), and that the age model for core U1361A is significantly different during the LIG from the original published version (which had limited constraints and assumed linear sedimentation rates throughout the late Pleistocene).

## Data availability

All data from this study can be found in the Supplementary Information and Supplementary Datasets 1–6.

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

## Acknowledgements

O.S. acknowledges support from the Japan Society for the Promotion of Science (JSPS) KAKENHI Grant Numbers 17H01166, 17H06318, and 20H00626. Mu.I. is supported by Grant-in-Aid for JSPS Fellows Grant Number 21J13181 and Sasagawa Scientific Research Grant from the Japan Society. Y.S. is supported by the JSPS KAKENHI Grant Numbers 16H05739, 17H06321, 19H00728. Ta. I. is supported by the JSPS KAKENHI Grant Numbers 21H01201. D.J.W. is supported by a Natural Environment Research Council independent research fellowship (NE/T011440/1). Thanks also go to N. Pratt for her kind technical support with detrital neodymium isotope measurements.

## Author contributions

O.S. designed the project. Mu.I., K.H., T.v.d.F., D.J.W., and O.S. conducted Nd isotope measurements. Y.S., M.H., and H.M. analysed Be isotopes. Mu.I. and To.I. measured the grain size. Mu.I. and Ta.I. measured the Radioralia assemblage. Mi.I. conducted the XRF analysis. All authors (Mu.I., O.S., D.J.W., Y.S., K.H., T.v.d.F., Mi.I, Ta.I., To.I., M.Y., M.H., H.M., and S.S.) interpreted the data and wrote the manuscript.

## Competing interests

The authors declare no competing interests.
