## [Peer Review File · Nature Communications]

Multiple episodes of ice loss from the Wilkes Subglacial Basin during the Last InterglacialReviewer #1 (Remarks to the Author):

In the manuscript "Multiple episodes of ice loss from the Wilkes Subglacial Basin during the last Interglacial" by M. Iizuka and others, the authors present a new high-resolution record (Be, Nd and IBRD) from the marine sediment core GC 1407 drilled offshore the Adelie Coast (Antarctica). Such record is expected to provide information regarding subglacial erosion (Nd), ice sheet melt episodes (Be), and iceberg discharge vents (IBRD) from the East Antarctic Ice Sheet. This new paleo-data, spanning the Holocene and the Last Interglacial Period (LIG), suggests that during the LIG the Wilkes Subglacial Basin (WSB) ice sheet went through two events of ice mass loss and ice margin retreat. Such record is also compared to the published data (Nd and IBRD) obtained from the sediment core record U1361A, collected offshore the WSB, and the $\delta^{18}O$ record from the TALDICE ice core, which is supposed to be sensitive to the WSB dynamics. All records are tuned on the AICC2012 age scale. The multi-proxy and multi-archive approach appears to confirm the hypothesis formulated by the authors. In particular, the U1361A sediment core record supports the two-step melting scenario of the WSB during the LIG. The $\delta^{18}O$ TALDICE record, which shows an increase during the late stages of the LIG, is interpreted as a drop in elevation at Talos Dome, related to an ice-sheet margin retreat in the WSB during the late stage of the LIG.

I find the new high-resolution records obtained from the GC 1407 sediment core to provide a very important picture of the past dynamics of the East Antarctic Ice Sheet (EAIS) since they are still poorly constrained. This new data set will certainly provide a useful addition in determining the EAIS sea level contribution during the LIG. The manuscript is generally clear and the enclosed results are worthy of publication, however I do suggest some improvements, especially regarding the interpretation of the results. In particular, I suggest improving the section regarding the TALDICE ice core $\delta^{18}O$ record interpretation and a broaden the discussion on the mechanism that may have caused the WSB ice sheet retreat to increase the scientific strength of the manuscript. Moreover, the SI file is rich in information that is necessary to be read to interpret the manuscript. For this reason, I think it is a pity to hide those useful notions in a separate file. I suggest shortening the sections in the SI when possible and concentrating on the main concepts and references in the manuscript. I am looking forward to reading the improved version of this manuscript.

Major comments on the main text

- Line 79: At this point of the introduction the authors introduce the proxies (Nd and Be) that are discussed in the manuscript. I suggest providing also a sentence on the IBRD record which is not mentioned here but is then discussed in the second part of the manuscript.

- Lines 116-119: Here Iizuka and co-authors introduce an essential point for the interpretation of the Nd record of the GC1047 core. I suggest providing references to support this statement, in particular regarding the ABW flow direction and the intensities changes during glacial/interglacial periods. This work might be an appropriate reference: Presti et al. (2011)

<https://doi.org/10.1016/j.margeo.2011.03.012>.

- Lines 116-132: This is a general comment on the paragraph as I am not myself an expert on sediment cores. If I understand correctly, Iizuka and co-authors suggest that the Nd values found in the GC1407 core during the LIG are interpreted as a mixture of detritus coming from the WSB area and the Adelie coast. The input from the Adelie coast is supposed to be continuous and to contribute to lowering the Nd values, while the two peaks are interpreted as margin retreat events in the WSB. I have some questions regarding this interpretation:

o Is there any previous evidence regarding the "continuous detrital input from the Adelie Coast" during the LIG? Could you please provide a reference? If this is a hypothesis of this study, could you please articulate it?

o At the end of the paragraph is stated that the Adelie Coast region was not affected by ice-sheet margin retreat during the LIG and this is supported by the modeling outputs from DeConto & Pollard (2016). I think this is an essential statement in this paragraph and it is the key to interpreting the new sediment record. For this reason, I suggest that the concept should be expanded a bit more and references need to be added to support

the statement (for both field data and modeling side). Regarding the modeling reference, I suggest adding other citations to the one of DeConto and Pollard (2016). Sutter et al (2020), for example, could be a good one. Also, Rignot et al (2019) indicate no recent retreat on the Adelie Coast.

- Lines 142-143: Please add references to support this statement. Valletta et al. (2018) cited some lines above that might be a good one.

- Lines 149-152: This sentence should be rephrased since it contains the term "changes" three times. Please specify what changes are you referring to (e.g. increase in Be ratios reflect ice sheet retreat etc).

- Figure 2: In the manuscript's main text the IBRD record of the GC1407 sediment core is described and compared to the one of the U1361A. Would be nice to see the IBRD record from the U1361A sediment core in Figure 2.

- Lines 153-156: Here only the IBRD record of the GC1407 is presented. Since the manuscript is mainly based on the comparison between the GC1407 and U1361A records, I think the text would benefit from the comparison of the two IBRD signals in order to strengthen the interpretation presented in this paper.

- Section "Assessment of ice mass loss in the WSB".

This section is, in my view, the core of the manuscript since the Nd and Be records from the GC1407 are compared to the TALDICE ice core $\delta^{18}\text{O}$ record during the LIG in order to estimate the WSB ice sheet retreat. However, I come across some inconsistencies. For this reason, I suggest that the section would largely improve if revised.

Here below some suggestions:

o Lines 162-165: This sentence is central to the discussion of this part of the manuscript. Please add the appropriate references regarding the possible interpretation of the $\delta^{18}\text{O}$ record. The isotopic oxygen ratios could be also sensitive to sea ice extension (Holloway et al., 2016). Regarding moisture sources, the air masses' origin changes are better pictured by the d-excess parameter rather than by the $\delta^{18}\text{O}$ itself.

o Explain better the connection between the increase in $\delta^{18}\text{O}$, the increase in local temperature (T), and the decrease in elevation. An explicative sentence is missing at the beginning of the section.

o Line 171-173: This statement comes too early in the section because the authors are still mentioning the hypothesis regarding the $\delta^{18}\text{O}$ signal interpretation. I suggest better exposing the hypothesis here since this part belongs to the introduction of the section.

The sentence could be placed, for example, at line 180, after ruling out the interpretation of the oxygen isotopic record.

o Lines 177-180: This statement is connected to the explanation provided in the SI. Please find my comments and suggestions on the SI below.

o In this part of the discussion, the TALDICE record is compared to the Be and Nd record from the GC1407 core. The IBRD record from the same core also provides interesting information, so it would be important to add it to the discussion to provide a more robust interpretation of the results.

o Lines 187-190: Iizuka and co-authors compute the surface elevation at Talos Dome using a lapse rate of $-0.53\text{‰}/100\text{m}$. This lapse rate is proposed by Sutter et al. (2020) and is inferred from present-day values, while Goursaud et al (2020) propose a lapse rate of $-0.93\text{‰}/100\text{m}$ modeled from LIG values. Please modify the citation on line 190. Probably it would be better to use the Goursaud et al. lapse rate for the elevation change calculation, but both lapse rates can be also used and the results can be compared. It would be also good to introduce some uncertainty in the elevation change calculation.

o Lines 190-197 and Figure 1: Iizuka and co-authors state that a 160 m elevation drop at Talos Dome would have caused a very large ice sheet retreat (pictured in Figure 1) as showed by simulations published by Sutter et al. (2020). I found this conclusion a bit unexpected because Sutter et al stated that, according to the comparison of the results of their different simulations "any retreat of the ice margin in the Wilkes and Aurora subglacial basins during the LIG must have been of limited extent". For this reason, I see this approach as very tricky because the authors are basing their conclusion on only one model simulation that was also discarded by the authors of the reference paper. I think this part of the discussion would largely benefit from a more detailed comparison

of model outputs from different publications to provide stronger support to their final statement. This comparison should also be visually included in Figure 1, showing different ice margin retreat from different papers.

- Line 207-208: Iizuka and co-authors here indicate that the causes of sea level rise during LIG remain unknown. This sentence is too broad because the paper focuses on the contribution to sea level increase of the WSB. For this reason, I suggest speculating on the cause that led to a dynamic WSB ice sheet during the LIG. Was it due to ocean warming (e.g. Chadwick et al. 2020 <https://doi.org/10.1016/j.quascirev.2019.106134>, Shukla et al. 2021 <https://doi.org/10.1029/2020GL090994>, Golledge et al. 2021 <https://doi.org/10.1029/2021GL094513>, Mengel and Levermann 2014) and/or atmospheric temperature increase during LIG (e.g. Golledge et al. 2017) or a combination of them? Provide appropriate references.

- Line 221: The authors here suggest that the Aurora and the Recovery Subglacial Basins are expected to be vulnerable to warming. Which kind of warming? Ocean warming or atmospheric warming? Both of them? The conclusion would benefit from a more extensive discussion on the retreat mechanism of the mentioned ice sheets with appropriate references.

Major Comments on Supplementary Information

- "Interpreting $\delta^{18}\text{O}$ changes in Talos Dome ice core": The interpretation of the TALDICE $\delta^{18}\text{O}$ record in the Supplementary Information needs to provide a robust understanding of the isotopic record. Here the discussion is very focused on references but I think adding a figure and some additional explanation would be worthy. Please find here some comments and suggestions:

Make sure that timings are described correctly, the authors refer to " $\delta^{18}\text{O}$ values after 126 ka" which could be confusing. I suggest indicating the timing window instead (e.g. 118-126 ka).

To improve the discussion regarding the ssNa+ record in TALDICE, I suggest it would be worth picturing it on a figure and comparing it also with the ssNa+ EDC record of the same period. How do the two records behave? Do they show any similarity/difference in correspondence of the late $\delta^{18}\text{O}$ increase in TALDICE? Does it support the initial interpretation?

Iizuka and co-authors state that the expansion of sea ice during the LIG would have caused a decrease in the TALDICE isotopic record due to the displacement of moisture sources. Could you quantify the decrease? Could you specify the displacement of moisture sources (latitude)?

The authors also add that changes in SST in moisture source areas cause only "small regional differences". Could you quantify this difference in terms of $\delta^{18}\text{O}$ (‰)?

Moreover, I also suggest reformulating the two concluding statement because the reference Goursaud et al. (2020) focus on the 128 ka isotopic peak and not on the late part of the LIG, so could not be used in this case; and the concluding remark should specify the "dynamic mechanism" affecting Talos Dome (e.g. site elevation decrease due to ice sheet retreat).

Minor comments

- Line 35-38: The sentence is too long. Please split it into two shorter sentences.

- Lines 51-54-56: Please substitute "multiple" with "different" or "several" (or a synonym)

- Line 56: Please change "which limits" with "limiting"

- Line 58-60: I think this sentence should center on the marine-based subglacial basins and their vulnerability. My suggestion is to focus only on the marine-based ice sheets in the EAIS and specify which kind of "small episodic changes" (e.g. warming SO) might have caused ice mass loss.

- Line 64: Please change the comma with a full stop. Rephrase the last sentence with a short conclusion, for example: "For this reason, a focus on the sedimentary record from the Antarctic margin is needed".

- Line 66: Please change "this" with "its".

- Line 96: The word "during" is repeated three times in two close sentences. Please change it with a synonym as "in correspondence of".

- Line 98: remove "during the LIG".

- Line 102: change "regarding" with "of".

- Line 160: Please change "to evaluate...LIG" with "To evaluate the extent of the WSB ice sheet retreat during the LIG".
 - Line 67-69: Please change the word order in the sentence as follows: "However, when and to what extent the ice-sheet margin in the WSB retreated during the LIG, compared to its present position, remains a matter of debate".
 - Lines 82-84: Please move this sentence to line 88, before the beginning of the sentence "The chronologies..". The description of Nd and Be isotopes should be close to the introduction on the marine sediment core to keep the paragraph easy to read.
 - Line 103: substitute "changes" with "variations".
 - Line 112: change "inferred" with "known".
 - Lines 156-158: Iizuka and co-authors state that the IBRD peak at the end of the LIG reflects an advance of the ice sheet margin rather than an ice melting episode. This sentence could be confusing. My suggestion is to change "end of the LIG" with "beginning to the glacial inception or onset of glaciation" specifying the time interval the authors are referring to.
 - Line 202: Please specify that the three scenarios are based on already published studies.
 - Figure 3: I think is very informative to add the three scenarios to this figure, however, I do suggest highlighting the fact that scenario number 3 is the one proposed also in this manuscript. Moreover, the figure would be more complete if also the IBRD record from the GC1407 sediment core is added. Then, depending on the hypothesis formulated in the new version of the manuscript, it would be interesting to add records of the forcing (ocean/atmosphere) that contributed to this scenario.
- Minor comments on Methods**
- Line 474: Change "were" with "when".
- Minor comments on Supplementary Information**
- Figures S2, S3, and S4: Those figures refer to the Method section and not to the Supplementary. The method section would be more complete if those figures are transferred to the Method part instead of being included in the Supplementary.
 - Figure S3: Please increase the resolution of the figure and use different markers for different records (e.g. diamonds, stars, squares, etc)
 - In the Supplementary Information please switch section 4 with section 5 because the Be isotopes record is discussed before the $\delta^{18}O$ TALDICE record in the manuscript. By doing this the SI gets easier to read.
 - Section 2 can be removed because such information is already available in the main text.

Reviewer #2 (Remarks to the Author):

In this study, changes in the Wilkes Subglacial Basin ice sheet during the Last Interglacial are examined using marine and ice core records. Changes in marine Nd and Be isotope signatures during the early and late phases of MIS 5e are interpreted as meltwater additions and discussed in the context of a WSB contribution to sea-level rise.

Given the lack of high-resolution marine records from the East Antarctic Ice Sheet for MIS 5e, I think this study is very important. The combination of eNd and Be is innovative and the overall interpretation of the data is sound. However, I wonder if the estimated WSB contribution of 0.5 m is significant enough to argue for scenarios 2 and 3 proposed for sea level rise during the LIG. Wouldn't Scenario 1 also account for such a small contribution? I am concerned that the mechanism underlying the retreat of the WSB ice sheet during the late LIG is not discussed in detail, which could help clarify whether it was a regional event or something that affected other parts of the East Antarctic ice sheet. Apart from these questions, which still need to be clarified, I consider this study worthy of publication in Nat. Commun.

A few minor comments that I think should also be addressed:

L84 and everywhere else: subscript Nd

L94: The stability of the eNd signal during the Holocene is inferred from only 5 samples in a time interval of ~ 3 ka. Due to this low resolution, episodic changes during the Holocene such as the one during early LIG (129-126 ka) might remain undetected.

L98: This statement is based on only one sample, which has a significantly higher radiogenic value (-8.5) than all other samples with values around -10.

L110: This statement is not supported by Figure 1, based on which 1) core U1361A is closer to the Adelie Coast than the George V Coast, and 2) the stronger influence of the Adelie Coast is supported by nearby core top signatures. Doesn't it make more sense here to argue with circulation?

L125-127: Couldn't changes in eNd values also be explained by changes in circulation? Based on Figure 1, there appear to be sufficient upstream sources of additional radiogenic signatures so that no ice sheet retreat needs to be assumed.

First of all, thank you very much for your positive response and useful comments on our manuscript. We have carefully considered all the reviewer's comments and have revised the manuscript, taking all the comments into account. Below are our responses to individual comments.

Reviewer #1 (Remarks to the Author):

In the manuscript “Multiple episodes of ice loss from the Wilkes Subglacial Basin during the last Interglacial” by M. Iizuka and others, the authors present a new high-resolution record (Be, Nd and IBRD) from the marine sediment core GC 1407 drilled offshore the Adelie Coast (Antarctica). Such record is expected to provide information regarding subglacial erosion (Nd), ice sheet melt episodes (Be), and iceberg discharge vents (IBRD) from the East Antarctic Ice Sheet. This new paleo-data, spanning the Holocene and the Last Interglacial Period (LIG), suggests that during the LIG the Wilkes Subglacial Basin (WSB) ice sheet went through two events of ice mass loss and ice margin retreat. Such record is also compared to the published data (Nd and IBRD) obtained from the sediment core record U1361A, collected offshore the WSB, and the $\delta^{18}\text{O}$ record from the TALDICE ice core, which is supposed to be sensitive to the WSB dynamics. All records are tuned on the AICC2012 age scale. The multi-proxy and multi-archive approach appears to confirm the hypothesis formulated by the authors. In particular, the U1361A sediment core record supports the two-step melting scenario of the WSB during the LIG. The $\delta^{18}\text{O}$ TALDICE record, which shows an increase during the late stages of the LIG, is interpreted as a drop in elevation at Talos Dome, related to an ice-sheet margin retreat in the WSB during the late stage of the LIG.

I find the new high-resolution records obtained from the GC 1407 sediment core to provide a very important picture of the past dynamics of the East Antarctic Ice Sheet (EAIS) since they are still poorly constrained. This new data set will certainly provide a useful addition in determining the EAIS sea level contribution during the LIG. The manuscript is generally clear and the enclosed results are worthy of publication, however I do suggest some improvements, especially regarding the interpretation of the results. In particular, I suggest improving the section regarding the TALDICE ice core $\delta^{18}\text{O}$ record interpretation and a broaden the discussion on the mechanism that may have caused the WSB ice sheet retreat to increase the scientific strength of the manuscript. Moreover, the SI file is rich in information that is necessary to be read to interpret the manuscript. For this reason, I think it is a pity to hide those useful notions in a separate file. I suggest shortening the sections in the SI when possible and concentrating on the main concepts and references in the manuscript. I am looking forward to reading the improved version of this manuscript.

Major comments on the main text

- Line 79: At this point of the introduction the authors introduce the proxies (Nd and Be) that are discussed in the manuscript. I suggest providing also a sentence on the IBRD record which is not mentioned here but is then discussed in the second part of the manuscript.

We have added a sentence introducing IBRD in the Introduction. (please see page 4 line 73)

- Lines 116-119: Here Iizuka and co-authors introduce an essential point for the interpretation of the Nd record of the GC1047 core. I suggest providing references to support this statement, in particular regarding the ABW flow direction and the intensities changes during glacial/interglacial periods.

This work might be an appropriate reference: Presti et al. (2011)

<https://doi.org/10.1016/j.margeo.2011.03.012>.

Thanks for this suggestion. We have cited the reference “Presti et al. (2011)”. (please see page 5 line 123)

- Lines 116-132: This is a general comment on the paragraph as I am not myself an expert on sediment cores. If I understand correctly, Iizuka and co-authors suggest that the Nd values found in the GC1407 core during the LIG are interpreted as a mixture of detritus coming from the WSB area and the Adelie coast. The input from the Adelie coast is supposed to be continuous and to contribute to lowering the Nd values, while the two peaks are interpreted as margin retreat events in the WSB. I have some questions regarding this interpretation:

o Is there any previous evidence regarding the “continuous detrital input from the Adelie Coast” during the LIG? Could you please provide a reference? If this is a hypothesis of this study, could you please articulate it?

Thank you for these useful suggestions. The sources of surface sediments have been investigated, and it has been indicated that the sediment of GC1407 is transported from upstream regions (e.g., Adelie Coast). However, direct evidence of “continuous detrital input from the Adelie Coast” during the LIG has not yet been obtained. Therefore, we have specified that this discussion is hypothetical in the revised manuscript. (please see page 5 lines 127–129).

o At the end of the paragraph is stated that the Adelie Coast region was not affected by ice-sheet margin retreat during the LIG and this is supported by the modeling outputs from DeConto & Pollard (2016). I think this is an essential statement in this paragraph and it is the key to interpreting the new sediment record. For this reason, I suggest that the concept should be expanded a bit more and

references need to be added to support the statement (for both field data and modeling side). Regarding the modeling reference, I suggest adding other citations to the one of DeConto and Pollard (2016). Sutter et al (2020), for example, could be a good one. Also, Rignot et al (2019) indicate no recent retreat on the Adelie Coast.

Thank you for the suggestions. We have added those references here. (please see page 6 lines 135–138)

- Lines 142-143: Please add references to support this statement. Valletta et al. (2018) cited some lines

We have added the reference. (please see page 6 line 147)

- Lines 149-152: This sentence should be rephrased since it contains the term “changes” three times. Please specify what changes are you referring to (e.g. increase in Be ratios reflect ice sheet retreat etc).

We have changed the word “change” with “retreat” to clearly indicate what the increase in Be isotopes reflects. (please see page 6 lines 157). We have kept the phrase “changes in the other process” and edited the other phrase to “shift in provenance” to improve readability.

- Figure 2: In the manuscript's main text the IBRD record of the GC1407 sediment core is described and compared to the one of the U1361A. Would be nice to see the IBRD record from the U1361A sediment core in Figure 2.

Thanks for this suggestion. We have added the IBRD record from U1361 in Figure 2.

- Lines 153-156: Here only the IBRD record of the GC1407 is presented. Since the manuscript is mainly based on the comparison between the GC1407 and U1361A records, I think the text would benefit from the comparison of the two IBRD signals in order to strengthen the interpretation presented in this paper.

We presented grain size of >1 mm as IBRD in the previous version. However, U1361 uses 250µm-2mm as IBRD. Therefore, for a more direct comparison between U1361 and GC1407, we now show 250µm-2mm as the IBRD in GC1407. Accordingly, we have updated the discussion, but this change does not affect our interpretation that the GC1407 and U1361 IBRD records are consistent. (please see pages 7 lines 160–166)

- Section “Assessment of ice mass loss in the WSB”.

This section is, in my view, the core of the manuscript since the Nd and Be records from the GC1407 are compared to the TALDICE ice core $\delta^{18}\text{O}$ record during the LIG in order to estimate the WSB ice sheet retreat. However, I come across some inconsistencies. For this reason, I suggest that the section would largely improve if revised.

Here below some suggestions:

o Lines 162-165: This sentence is central to the discussion of this part of the manuscript. Please add the appropriate references regarding the possible interpretation of the $\delta^{18}\text{O}$ record. The isotopic oxygen ratios could be also sensitive to sea ice extension (Holloway et al., 2016). Regarding moisture sources, the air masses' origin changes are better pictured by the d-excess parameter rather than by the $\delta^{18}\text{O}$ itself.

We have added the reference the reviewer suggested. (please see page 7 line 171)

o Explain better the connection between the increase in $\delta^{18}\text{O}$, the increase in local temperature (T), and the decrease in elevation. An explicative sentence is missing at the beginning of the section.

We have revised the sentence "change in temperature" and "change in ice sheet elevation" to "temperature increase" and "elevation drop", respectively, to make it clearer. (please see page 7 lines 168–179)

o Line 171-173: This statement comes too early in the section because the authors are still mentioning the hypothesis regarding the $\delta^{18}\text{O}$ signal interpretation. I suggest better exposing the hypothesis here since this part belongs to the introduction of the section.

The sentence could be placed, for example, at line 180, after ruling out the interpretation of the oxygen isotopic record.

We have modified this sentence so that we are not yet interpreting the changes as necessarily driven by ice-sheet elevation change and ice-sheet retreat. (please see page 7 lines 178–179)

o Lines 177-180: This statement is connected to the explanation provided in the SI. Please find my comments and suggestions on the SI below.

o In this part of the discussion, the TALDICE record is compared to the Be and Nd record from the GC1407 core. The IBRD record from the same core also provides interesting information, so it would be important to add it to the discussion to provide a more robust interpretation of the results.

Thank you for the useful suggestions. We agree with your point that comparing IBRD with the other

records can provide supporting evidence. Since we already discussed the comparison between ϵ_{Nd} , Be, and IBRD in the previous section (please pages 6 lines 160–166), we refrained from discussing IBRD together with the ice core $\delta^{18}O$ records in this section.

o Lines 187-190: Iizuka and co-authors compute the surface elevation at Talos Dome using a lapse rate of $-0.53\%/100\text{m}$. This lapse rate is proposed by Sutter et al. (2020) and is inferred from present-day values, while Goursaud et al (2020) propose a lapse rate of $-0.93\%/100\text{m}$ modeled from LIG values. Please modify the citation on line 190. Probably it would be better to use the Goursaud et al. lapse rate for the elevation change calculation, but both lapse rates can be also used and the results can be compared. It would be also good to introduce some uncertainty in the elevation change calculation.

Thank you for this important suggestion. We have added new discussion using three different proposed lapse rates. The different values lead to a range of estimates for elevation change, which provides an indication of the level of uncertainty associated with the approach. (please see page 8 lines 197–206)

o Lines 190-197 and Figure 1: Iizuka and co-authors state that a 160 m elevation drop at Talos Dome would have caused a very large ice sheet retreat (pictured in Figure 1) as showed by simulations published by Sutter et al. (2020). I found this conclusion a bit unexpected because Sutter et al stated that, according to the comparison of the results of their different simulations “any retreat of the ice margin in the Wilkes and Aurora subglacial basins during the LIG must have been of limited extent”. For this reason, I see this approach as very tricky because the authors are basing their conclusion on only one model simulation that was also discarded by the authors of the reference paper. I think this part of the discussion would largely benefit from a more detailed comparison of model outputs from different publications to provide stronger support to their final statement. This comparison should also be visually included in Figure 1, showing different ice margin retreat from different papers.

As the reviewer explains, Sutter et al. (2020) propose that the ice-sheet retreat in the WSB during the LIG was limited, which appears to be different from our interpretation. However, Sutter et al. (2020) focused on the changes in the ice sheet at 128 ka and did not discuss the late LIG interval. We applied the ice sheet model result discussed above to help interpret the ice sheet elevation drop during the late LIG.

- Line 207-208: Iizuka and co-authors here indicate that the causes of sea level rise during LIG remain unknown. This sentence is too broad because the paper focuses on the contribution to sea level increase of the WSB. For this reason, I suggest speculating on the cause that led to a dynamic

WSB ice sheet during the LIG. Was it due to ocean warming (e.g. Chadwick et al. 2020 <https://doi.org/10.1016/j.quascirev.2019.106134>, Shukla et al. 2021 <https://doi.org/10.1029/2020GL090994>, Golledge et al. 2021 <https://doi.org/10.1029/2021GL094513>, Mengel and Levermann 2014) and/or atmospheric temperature increase during LIG (e.g. Golledge et al. 2017) or a combination of them? Provide appropriate references.

We appreciate your comments, and thank you for pointing us to those papers regarding the oceanic forcing. In accordance with the comment, we now discuss the cause of the change in the ice sheet in the WSB during the LIG. Although we cannot be definitive due to the limited paleoceanographic records in the high-latitude Southern Ocean, we discuss the possibility that ocean warming may have played an important role in causing ice sheet mass loss in the late LIG. (please see pages 8–9 lines 214–238)

- Line 221: The authors here suggest that the Aurora and the Recovery Subglacial Basins are expected to be vulnerable to warming. Which kind of warming? Ocean warming or atmospheric warming? Both of them? The conclusion would benefit from a more extensive discussion on the retreat mechanism of the mentioned ice sheets with appropriate references.

The Aurora and the Recovery Subglacial Basins are considered to be susceptible to ocean warming. We have added a sentence explaining this point, including two appropriate references. (please see page 10 line 265)

Major Comments on Supplementary Information

- “Interpreting $\delta^{18}\text{O}$ changes in Talos Dome ice core”: The interpretation of the TALDICE $\delta^{18}\text{O}$ record in the Supplementary Information needs to provide a robust understanding of the isotopic record. Here the discussion is very focused on references but I think adding a figure and some additional explanation would be worthy. Please find here some comments and suggestions:

Make sure that timings are described correctly, the authors refer to “ $\delta^{18}\text{O}$ values after 126 ka” which could be confusing. I suggest indicating the timing window instead (e.g. 118-126 ka).

When we first submitted this manuscript, it was harder to fully interpret the TALDICE $\delta^{18}\text{O}$ record because the d-excess data from TALDICE for the LIG was not available. However, a paper has recently been published that included those data and discussed its interpretation (Crotti et al., 2022). In their

study, Crotti et al. (2022) compared the deuterium excess ($d\text{-excess} = \delta D - 8 \cdot \delta^{18}O$) and $ssNa^+$ records from EDC and TALDICE during the LIG, and concluded that the different signals in the EDC and TALDICE $\delta^{18}O$ records could not be explained by processes such as regional sea-ice changes and should instead reflect changes in the ice-sheet elevation at TALDICE. For this reason, we reference this paper and have reduced our own discussion that previously attempted to interpret the TALDICE $\delta^{18}O$ record.

To improve the discussion regarding the $ssNa^+$ record in TALDICE, I suggest it would be worth picturing it on a figure and comparing it also with the $ssNa^+$ EDC record of the same period. How do the two records behave? Do they show any similarity/difference in correspondence of the late $\delta^{18}O$ increase in TALDICE? Does it support the initial interpretation? Iizuka and co-authors state that the expansion of sea ice during the LIG would have caused a decrease in the TALDICE isotopic record due to the displacement of moisture sources. Could you quantify the decrease? Could you specify the displacement of moisture sources (latitude)?

Crotti et al. (2022) discussed possible changes in $\delta^{18}O$ values due to sea ice, including a comparison of the $ssNa^+$ records in EDC and TALDICE (their Fig. 2). The $ssNa^+$ records in both cores vary in parallel during the LIG, suggesting that the impact of sea ice changes on the $\delta^{18}O$ records is probably limited, and certainly indicating that spatial differences in sea ice changes between the source regions supplying the two cores could not explain the discrepancy in their $\delta^{18}O$ records. In the original manuscript, we discussed this point based only on the $ssNa^+$ record from TALDICE. With publication of the complete records including $d\text{-excess}$ (Crotti et al., 2022), there is now further support for a negligible role of sea ice changes. Here, we cite the detailed work of Crotti et al. (2022) for this point. However, we do note that it remains a challenge to fully quantify the impact of sea ice.

The authors also add that changes in SST in moisture source areas cause only “small regional differences”. Could you quantify this difference in terms of $\delta^{18}O$ (‰)?

We discussed the effect of SST changes on $\delta^{18}O$ values in the original version of our manuscript, but the effect has now been discussed in detail by Crotti et al. (2022) (published after we submitted our manuscript), and they concluded that the effects is negligible. Therefore, in the revised version, we have addressed this question by citing their study.

Moreover, I also suggest reformulating the two concluding statement because the reference Goursaud et al. (2020) focus on the 128 ka isotopic peak and not on the late part of the LIG, so could not be used in this case; and the concluding remark should specify the “dynamic mechanism” affecting Talos

Dome (e.g. site elevation decrease due to ice sheet retreat).

We agree with your comment that the Goursaud et al. (2020) model at 128 ka is probably not directly comparable to the scenario we are considering during the late LIG. However, regional air temperature change during the late LIG (122–118 ka) remains somewhat uncertain. However, the difference in temperature between the TALDICE and EDC not only at 128 ka but also at the LGM and the present is less than 0.5°C (equivalent to a change in $\delta^{18}O$ of less than 0.3‰.) (e.g., Buizert et al., 2021; Jun et al., 2020). Despite that uncertainty, it indicates that the impact of local air temperature increases on the $\delta^{18}O$ records is small even during the late LIG. Furthermore, since such a discussion is not involved in previous studies on the relationship between $\delta^{18}O$ and ice sheet elevation, we have removed it to simplify the content.

Minor comments

- Line 35-38: The sentence is too long. Please split it into two shorter sentences.

We have revised the sentence (please see page 2 lines 26–30)

- Lines 51-54-56: Please substitute “multiple” with “different” or “several” (or a synonym)

We have revised the word to reduce repetition of ‘several’ (please see page 3 line 42)

- Line 56: Please change “which limits” with “limiting”

We have revised the word (please see page 3 line 48)

- Line 58-60: I think this sentence should center on the marine-based subglacial basins and their vulnerability. My suggestion is to focus only on the marine-based ice sheets in the EAIS and specify which kind of “small episodic changes” (e.g. warming SO) might have caused ice mass loss.

We have revised the sentence according to these suggestions (please see page 3 lines 51–52)

- Line 64: Please change the comma with a full stop. Rephrase the last sentence with a short conclusion, for example: “For this reason, a focus on the sedimentary record from the Antarctic margin is needed”.

Thanks for this nice suggestion! We have revised the sentence (please see page 3 lines 57–58)

- Line 66: Please change “this” with “its”.

We have revised the sentence (please see pages 3 line 62)

- Line 96: The word “during” is repeated three times in two close sentences. Please change it with a synonym as “in correspondence of “.

We have replaced the word “during” with “within” because it represents a period. (please see page 4 lines 92)

- Line 98: remove “during the LIG”.

We have removed the word “during the LIG”.

- Line 102: change “regarding” with “of”.

We have changed the word “regarding” with “of”. (please see pages 5 lines 100)

- Line 160: Please change “to evaluate...LIG” with “To evaluate the extent of the WSB ice sheet retreat during the LIG”.

We have revised the sentence in accordance with the comment (please see page 7 line 168)

- Line 67-69: Please change the word order in the sentence as follows: “However, when and to what extent the ice-sheet margin in the WSB retreated during the LIG, compared to its present position, remains a matter of debate”.

We have revised the sentence according to the comment (please see page 3 lines 60–62)

- Lines 82-84: Please move this sentence to line 88, before the beginning of the sentence “The chronologies..”. The description of Nd and Be isotopes should be close to the introduction on the marine sediment core to keep the paragraph easy to read.

We have moved this sentence in accordance with the comment (please see page 4 lines 80–82)

- Line 103: substitute “changes” with “variations”.

We have revised the word “changes” with “variations” (please see page 5 lines 102)

- Line 112: change “inferred” with “known”.

Thanks for the suggestion. However, we would like to keep “inferred” because this insight is derived from aerogeophysical surveys, not from direct evidence. Hence, we believe that “infer” is more appropriate in this case.

- Lines 156-158: Iizuka and co-authors state that the IBRD peak at the end of the LIG reflects an advance of the ice sheet margin rather than an ice melting episode. This sentence could be confusing. My suggestion is to change “end of the LIG” with “beginning to the glacial inception or onset of glaciation” specifying the time interval the authors are referring to.

Thank you for this helpful comment. We have removed the potentially confusing sentence in the revised manuscript.

- Line 202: Please specify that the three scenarios are based on already published studies.

We cited the appropriate references here to clarify that the scenarios have been proposed by a range of other authors in previous studies (please see page 9 line 242).

- Figure 3: I think is very informative to add the three scenarios to this figure, however, I do suggest highlighting the fact that scenario number 3 is the one proposed also in this manuscript.

We appreciate this comment, but do not feel we should yet rule out the other scenarios. In providing the first ice-proximal evidence supporting an Antarctic origin for global sea-level rise during the late LIG, we support scenarios 2 and 3 (which both have a late LIG sea-level peak). However, we did not reject scenario 1, because our records (IBRD, Be isotopes, Nd isotopes) also show a peak during the early LIG. Therefore, we did not highlight scenario 3 in the figure, and prefer to leave this discussion to the main text.

Moreover, the figure would be more complete if also the IBRD record from the GC1407 sediment core is added. Then, depending on the hypothesis formulated in the new version of the manuscript, it would be interesting to add records of the forcing (ocean/atmosphere) that contributed to this scenario.

We have added the IBRD record in Figure 3 and Figure 4, as suggested. Please also note that Figure 3 is newly-added and enables a comparison with records of the ocean-atmosphere forcing, enabling us to discuss the cause of ice-sheet mass loss during the LIG.

Minor comments on Methods

- Line 474: Change “were” with “when”.

We have replaced with ‘when’ as suggested .

Minor comments on Supplementary Information

- Figures S2, S3, and S4: Those figures refer to the Method section and not to the Supplementary. The method section would be more complete if those figures are transferred to the Method part instead of being included in the Supplementary.

We agree and have transferred the figures to the Method part as suggested.

- Figure S3: Please increase the resolution of the figure and use different markers for different records (e.g. diamonds, stars, squares, etc)

We have revised these figures to update the symbols and increase the resolution (now Figures 7 and 8) with high resolution (please see Figures 7, 8)

- In the Supplementary Information please switch section 4 with section 5 because the Be isotopes record is discussed before the $\delta^{18}\text{O}$ TALDICE record in the manuscript. By doing this the SI gets easier to read.

We have moved the Supplementary Information to the Methods section in the main text and have re-ordered those sections.

- Section 2 can be removed because such information is already available in the main text.

We have removed this section.

Reviewer #2 (Remarks to the Author):

In this study, changes in the Wilkes Subglacial Basin ice sheet during the Last Interglacial are examined using marine and ice core records. Changes in marine Nd and Be isotope signatures during the early and late phases of MIS 5e are interpreted as meltwater additions and discussed in the context of a WSB contribution to sea-level rise.

Given the lack of high-resolution marine records from the East Antarctic Ice Sheet for MIS 5e, I think this study is very important. The combination of eNd and Be is innovative and the overall interpretation of the data is sound.

However, I wonder if the estimated WSB contribution of 0.5 m is significant enough to argue for scenarios 2 and 3 proposed for sea level rise during the LIG. Wouldn't Scenario 1 also account for such a small contribution?

We agree that only mass loss of the ice sheet in the WSB, if restricted to 0.5 m, would not be enough by itself to explain the magnitude of sea level rise during the late LIG in Scenarios 2 and 3. For this reason, in the last paragraph we discuss the possibility of mass loss in other basins of East Antarctica, such as the Aurora and Recovery Basins. Those basins are likely to behave similarly to the WSB in response to ocean warming, or may even be more sensitive (e.g. Golledge et al., 2017, GRL). (please see page 9 lines 260–265). Also, as you point out, the ice mass loss of the WSB (~0.5 m) may have contributed to the elevated sea levels during the late LIG in scenario 1, so our new record does not in itself rule out Scenario 1. (please see page 9 lines 251–254).

I am concerned that the mechanism underlying the retreat of the WSB ice sheet during the late LIG is not discussed in detail, which could help clarify whether it was a regional event or something that affected other parts of the East Antarctic ice sheet.

We agree with the reviewer that a discussion on the possible mechanisms of retreat was definitely warranted, and would serve to strengthen the paper. Therefore, we have added a discussion of the mechanisms underlying the retreat of the WSB ice sheet during the late LIG. At this stage it is hard to be definitive, due to the limited paleoceanographic records in the high-latitude Southern Ocean. However, the existing evidence (from both proxy-based records of ocean temperature and ice-sheet modelling) appears to support that ocean warming may have played an important role (and was likely more viable than atmospheric warming) in causing ice loss from the WSB during the late LIG. (please see pages 8–9 lines 214–238, and the newly-added Figure 3)

A few minor comments that I think should also be addressed:

L84 and everywhere else: subscript Nd

We are sorry for the incorrect notation, which we have now corrected (please see page 4 line 76, and elsewhere).

L94: The stability of the eNd signal during the Holocene is inferred from only 5 samples in a time interval of ~3 ka. Due to this low resolution, episodic changes during the Holocene such as the one during early LIG (129-126 ka) might remain undetected.

We agree with the reviewer, and have removed the word “stable”. (please see page 4 line 89).

L98: This statement is based on only one sample, which has a significantly higher radiogenic value (-8.5) than all other samples with values around -10.

This is a fair point. Indeed, one of the reasons for generating the high-resolution record in core GC1407 was to verify if those peaks in U1361 are significant or not. Our high-resolution records in GC1407 clearly show the presence of two peaks, and are consistent with the timing in the low-resolution record from core U1361A. We have added a sentence to address this point in the revised manuscript. (please see page 4 lines 95–97)

L110: This statement is not supported by Figure 1, based on which 1) core U1361A is closer to the Adelie Coast than the George V Coast, and 2) the stronger influence of the Adelie Coast is supported by nearby core top signatures. Doesn't it make more sense here to argue with circulation?

We agree with the reviewer. We view the downslope transport via turbidity currents as necessary to supply the sediment from the Antarctic shelf to offshore (i.e. transport perpendicular to the margin), but we expect it to be further transported to the west by deep-ocean currents. In the revised manuscript, we argue that transport by ocean circulation helps explain the similarity between the two records. (please see page 5 lines 105)

L125-127: Couldn't changes in eNd values also be explained by changes in circulation? Based on Figure 1, there appear to be sufficient upstream sources of additional radiogenic signatures so that no ice sheet retreat needs to be assumed.

In the Methods section, we describe the effect of changes in ocean circulation (bottom currents) on

the detrital ϵ_{Nd} record, based on comparison to a record of the mean sortable silt grain size (SS: 10–63 μ m), which is an indicator of the velocity of the bottom current. We find that the SS does not correlate with the ϵ_{Nd} record. This observation suggests that the ϵ_{Nd} record primarily reflects changes in sediment inputs related to ice margin changes upstream in the WSB region, and cannot be explained by changes only in ocean circulation such as bottom current speed. (please see pages 22–23 lines 513–531)

Reviewer #1 (Remarks to the Author):

This is my second review of the manuscript of Iizuka and co-authors. I am glad that the authors have exhaustively addressed suggestions provided by both reviewers and modified the manuscript accordingly. I believe that the manuscript is now worth publication. However, I still have some minor comments that should be addressed to draft the final version of the paper:

- Line 66: I suggest adding the reference of Crotti et al. 2022 when talking about LIG since the interpretation of the GC1407 marine sediment core record is now interpreted on the basis of the findings presented in cited the paper.
- Lines 80-82: This sentence explains how the ice core and sediment core chronologies were tuned. It would be better to move this sentence to the end of the paragraph, after the sentence that introduces the investigation of Antarctic ice core records, to improve the coherency of the text.
- Line 167: This paragraph now explains how to interpret the TALDICE isotopic record during the LIG and to evaluate the extent of the WSB ice sheet retreat. The revision of the paragraph provides a more accurate interpretation in comparison to the previous version of the "Interpreting $\delta^{18}O$ changes in Talos Dome ice core" section included in the Methods. This improvement is connected, as stated by the authors, to the recent publication of a paper by Crotti et al. which provides a picture of the WSB retreat based on the interpretation of the TALDICE isotopic record and the comparison with U1361A marine core record and ice sheet model simulations. Since the discussion regarding the ice mass loss in the WSB is now based on this published paper, I think this paragraph should be shortened. My suggestion is to simplify the text between lines 168 and 196 because most of this discussion is already presented in Crotti et al. It would be better to cite the paper and the main findings which are useful to interpret the GC1407 marine core record, which is the real novelty presented in this manuscript. Some additional lines focusing on the comparison of GC1407, U1361A and TALDICE record would be certainly interesting to be read.
- Lines 197-207: This paragraph should be removed because it reproduces the same hypotheses presented in Crotti et al. A simple sentence referring to the elevation changes calculated by Crotti et al should be enough.
- Lines 207-213: In this interpretation of the $\delta^{18}O$ TALDICE record with ice sheet simulations it would be good to add a sentence also on the ice sheet simulation performed in Crotti et al since you are citing their work and compare it to the results obtained by Sutter et al. 2020.
- Line 228: please change "ice-sheld and ice-sheet melting" with grounding line inland migration.
- Line 238: Add "during the LIG" at the end of the sentence.
- Line 242: Please substitute "during the LIG" with "for this interglacial period" to avoid repetition.
- Line 247: Please change "to elevated global sea levels" with "to global sea level increase".
- Line 249: The term "ice-proximal" is a bit unclear. Could you please substitute it?
- Line 255: I would add also the reference to Crotti et al because they also performed model simulations.

Reviewer #1 (Remarks to the Author):

This is my second review of the manuscript of Iizuka and co-authors. I am glad that the authors have exhaustively addressed suggestions provided by both reviewers and modified the manuscript accordingly. I believe that the manuscript is now worth publication. However, I still have some minor comments that should be addressed to draft the final version of the paper:

We appreciate the time and effort that you and each of the reviewers have dedicated to providing insightful feedback to help strengthen our discussion. We have now incorporated further changes that reflect your suggestions and comments on the revised version, as detailed in the following point-by-point response.

- Line 66: I suggest adding the reference of Crotti et al. 2022 when talking about LIG since the interpretation of the GC1407 marine sediment core record is now interpreted on the basis of the findings presented in cited the paper.

We have cited the reference of Crotti et al. (2022) in line 68 and elsewhere (please see page 3, line 68)

- Lines 80-82: This sentence explains how the ice core and sediment core chronologies were tuned. It would be better to move this sentence to the end of the paragraph, after the sentence that introduces the investigation of Antarctic ice core records, to improve the coherency of the text.

We have moved this sentence to the end of the paragraph (please see page 4, lines 85–87).

- Line 167: This paragraph now explains how to interpret the TALDICE isotopic record during the LIG and to evaluate the extent of the WSB ice sheet retreat. The revision of the paragraph provides a more accurate interpretation in comparison to the previous version of the “Interpreting $\delta^{18}\text{O}$ changes in Talos Dome ice core” section included in the Methods. This improvement is connected, as stated by the authors, to the recent publication of a paper by Crotti et al. which provides a picture of the WSB retreat based on the interpretation of the TALDICE isotopic record and the comparison with U1361A marine core record and ice sheet model simulations. Since the discussion regarding the ice mass loss in the WSB is now based on this published paper, I think this paragraph should be shortened. My suggestion is to simplify the text between lines 168 and 196 because most of this discussion is already presented in Crotti et al. It would be better to cite the paper and the main findings which are useful to interpret the GC1407 marine core record, which is the real novelty presented in this manuscript. Some additional lines focusing on the comparison of GC1407, U1361A and TALDICE record would be certainly interesting to be read.

We have simplified the text in this section by citing Crotti et al. (2022) (please see page 7, lines 171–181).

- Lines 197-207: This paragraph should be removed because it reproduces the same hypotheses presented in Crotti et al. A simple sentence referring to the elevation changes calculated by Crotti et al should be enough.

We have removed the paragraph.

- Lines 207-213: In this interpretation of the $\delta^{18}\text{O}$ TALDICE record with ice sheet simulations it would be good to add a sentence also on the ice sheet simulation performed in Crotti et al since you are citing their work and compare it to the results obtained by Sutter et al. 2020.

We have added a sentence on the ice sheet simulation by Crotti et al (2022) (please see page 8, lines 202–205).

- Line 228: please change “ice-sheld and ice-sheet melting” with grounding line inland migration.

We have replaced “ice-shelf and ice-sheet melting” with “inland migration of the grounding line” (please see page 9, line 222).

- Line 238: Add “during the LIG” at the end of the sentence.

We have added “during the LIG” at the end of the sentence (please see page 9, lines 232–233).

- Line 242: Please substitute “during the LIG” with “for this interglacial period” to avoid repetition.

We have substituted “during the LIG” with “for this interglacial period” (please see page 9, lines 236–237).

- Line 247: Please change “to elevated global sea levels” with “to global sea level increase”.

We have changed “to elevated global sea levels” with “to global sea level increase” (please see page 9, line 242).

- Line 249: The term “ice-proximal” is a bit unclear. Could you please substitute it?

We have removed “ice-proximal” and changed the text to “Our study provides evidence for ice mass loss from the WSB during the late LIG, thereby supporting an Antarctic origin for global sea-level rise at this time” (please see page 9, lines 244–245).

- Line 255: I would add also the reference to Crotti et al because they also performed model simulations.

We have added the reference to Crotti et al. (2022) (please see page 9, line 250).